# Allosteric cross-talk in chromatin can mediate drug-drug synergy

Zenita Adhireksan[1,*], Giulia Palermo[2,*], Tina Riedel[2,*], Zhujun Ma[1,*], Reyhan Muhammad[1], Ursula Rothlisberger[2], Paul J. Dyson[2] & Curt A. Davey[1,3]

Exploitation of drug–drug synergism and allostery could yield superior therapies by capitalizing on the immensely diverse, but highly specific, potential associated with the biological macromolecular landscape. Here we describe a drug–drug synergy mediated by allosteric cross-talk in chromatin, whereby the binding of one drug alters the activity of the second. We found two unrelated drugs, RAPTA-T and auranofin, that yield a synergistic activity in killing cancer cells, which coincides with a substantially greater number of chromatin adducts formed by one of the compounds when adducts from the other agent are also present. We show that this occurs through an allosteric mechanism within the nucleosome, whereby defined histone adducts of one drug promote reaction of the other drug at a distant, specific histone site. This opens up possibilities for epigenetic targeting and suggests that allosteric modulation in nucleosomes may have biological relevance and potential for therapeutic interventions.

[1] School of Biological Sciences, Nanyang Technological University, 60 Nanyang Drive, Singapore 637551, Singapore. [2] Institut des Sciences et Ingénierie Chimiques, Ecole Polytechnique Fédérale de Lausanne (EPFL), CH-1015 Lausanne, Switzerland. [3] NTU Institute of Structural Biology, Nanyang Technological University, 59 Nanyang Drive, Singapore 636921, Singapore. * These authors contributed equally to this work. Correspondence and requests for materials should be addressed to U.R. (email: ursula.roethlisberger@epfl.ch) or to P.J.D. (email: paul.dyson@epfl.ch) or to C.A.D. (email: davey@ntu.edu.sg).

Therapies based on an individual drug frequently have limited efficacy with poor resistance and safety profiles[1]. This arises largely from the network basis of cell structure and activity, whereby numerous interacting pathways can potentially allow for compensation, counteraction or neutralization of the initial drug effect. To circumvent these shortcomings of single agent regimens, much effort has been focusing on discovering drug combinations that function in a therapeutically productive fashion. Combinations that act in synergy can be especially advantageous by lowering therapeutic dose, thereby diminishing off-site targeting and corresponding side effects. In fact, synergistic drug combinations have been shown to generally coincide with improved therapeutically specific selectivity[2].

The origin of a particular drug–drug synergy can be varied, but commonly comes about via one of several general phenomena, including complementing or facilitating actions or effects that antagonize counteraction[1]. The molecular mechanism of the synergy may thus arise from targeting different molecules that do or do not interact over the course of the therapeutic event. Alternatively, targeting of two different sites on a given molecule or molecular assembly may underlie the synergy, which raises the possibility that the drug–drug interaction is mediated allosterically. In this scenario, there is no direct interaction between the agents, but rather the binding of one drug shifts the dynamic equilibrium of conformational substates in such a way that it influences the association of the other drug[3].

Largely because of the enormously diverse binding site and conformational landscape available to allosteric modulation, it has been argued that the discovery of allosterically acting drugs has special potential for developing innovative therapies[4]. But the challenge is first to uncover these compounds and second to define the mechanism of action, allowing for rational design strategies[5]. There are numerous examples of well-defined allosteric drug mechanisms behind protein signalling systems, in particular G protein-coupled receptors[6] and different kinases[7], and also for a variety of drug metabolizing enzymes, the P450 cytochromes[8]. However, instances of allosteric drug activities on the histone protein-packaged form of cellular DNA, chromatin, are not clear, and chromatin drug–drug interactions of allosteric origin have not been reported.

Here we characterize a synergistic impact on tumour cells mediated by RAPTA-T, [($\eta^6$-$p$-toluene)Ru(1,3,5-triaza-7-phosphaa-damantane)Cl$_2$], and auranofin, [3,4,5-triacetyloxy-6-acetyloxy-methyl,oxane-2-thiolate)Au(triethylphosphanium)] (AUF; Fig. 1a). RAPTA-T is a bifunctional ruthenium agent that displays low toxicity and is proficient at inhibiting both primary tumour growth and the spreading and growth of solid metastatic tumours in mice[9,10]. This agent also has anti-angiogenic properties that likely contribute to the antimetastasis activities[11]. AUF, on the other hand, is a monofunctional gold compound having high cytotoxicity, which was developed more than 30 years ago as an orally administrable drug to treat rheumatoid arthritis[12]. However, recent years have witnessed a resurgence of interest in AUF as a potential anticancer or antimicrobial agent, since it has come to light that it can inhibit both inflammatory pathways and thiol redox enzymes[13].

The cytotoxic effect of AUF towards cancer cells appears to stem, at least in part, from targeting protein factors involved in pro-inflammatory pathways that facilitate tumour growth and development[13]. Although DNA has to our knowledge not been reported as an AUF target, we had previously discovered that certain metal-based compounds can form site-specific adducts on the histone proteins that package DNA into the basic repeating unit of chromatin, the nucleosome[14–17]. We therefore investigated this as a possibility for AUF targeting here and found that AUF and RAPTA-T both form histone protein adducts in the nucleosome core, albeit at distantly related sites. Moreover, the presence of RAPTA-T substantially facilitates AUF adduct formation. This synergism elicited by RAPTA-T and AUF appears to be mediated via an allosteric mechanism within the nucleosome, which has biological implications and may open up new avenues for drug development beyond metal-based drugs.

## Results

**Tumour cell cytotoxicity synergism with RAPTA-T and AUF.** RAPTA-T has sufficiently low toxicity that treatment of human ovarian cancer A2780 cells with a 30 μM concentration of the compound yields no measurable impact on cell viability, whereas the corresponding IC$_{50}$ value for AUF is $<100$ nM (Fig. 1b). However, when cells are treated with both RAPTA-T and AUF, cell viability is reduced significantly relative to treatment with AUF alone (Fig. 1b). The drug combination coincides with a synergistic impact on viability, which is maximal (over the concentration ranges tested) at approximately 62 μM RAPTA-T (Fig. 1c). In addition, we found that there is a significantly greater reduction in viability when cells are first pre-treated with RAPTA-T followed by incubation with AUF as compared to vice versa—pre-treatment with AUF, followed by incubation with RAPTA-T (Supplementary Fig. 1). This indicates that RAPTA-T activity has a sensitizing effect on the tumour cells towards AUF.

**Synergistic accumulation of cellular chromatin drug adducts.** We next employed inductively coupled plasma mass spectrometry (ICP-MS) to quantify drug association with chromatin in tumour cells. Treatment of cells with either RAPTA-T alone or AUF alone resulted in substantial levels of chromatin-metal adduct formation, $139 \pm 29$ pmol per μg of DNA of ruthenium and about twice that level of gold, $345 \pm 138$ pmol per μg of DNA, for the two individual compounds (Fig. 1d). When cells were subsequently co-treated with RAPTA-T and AUF together, the level of chromatin-bound ruthenium remained approximately the same, to within experimental error, relative to the treatment with RAPTA-T alone. However, this co-treatment yielded a nearly threefold greater quantity, $1,006 \pm 324$ pmol per μg of DNA, of chromatin-bound gold relative to AUF treatment alone, indicating that RAPTA-T promotes chromatin-AUF adduct formation.

**RAPTA-T forms specific histone adducts on the nucleosome.** To shed light on the chromatin site selectivity of RAPTA-T, we treated nucleosome core particle (NCP) crystals with the compound and characterized adduct formation by X-ray crystallography (Supplementary Table 1; Supplementary Fig. 2). RAPTA-T generates adducts, via substitution of the chloride ligands, at two adjacent sites on the face of the nucleosome core where a preponderance of glutamate and aspartate residues from the histone H2A–H2B dimer form an 'acidic patch'[14–18], known to be a key protein binding motif for chromatin regulation[19–21] (Fig. 2). The two RAPTA-T binding sites, RU1 and RU2, entail coordination of the ruthenium centre to the H2A E61 and E64 carboxylate groups (RU1) and the imidazole and carboxylate groups of H2B H106 and E102 (RU2). The close proximity of the two sites coincides with a van der Waals (hydrophobic) contact between the toluene ring of RU1 and the PTA ligand of RU2.

**Distinct histone site reactivity of AUF with RAPTA-T present.** We conducted NCP crystal treatment trials with AUF using the same approach as for the derivatization with RAPTA compounds.

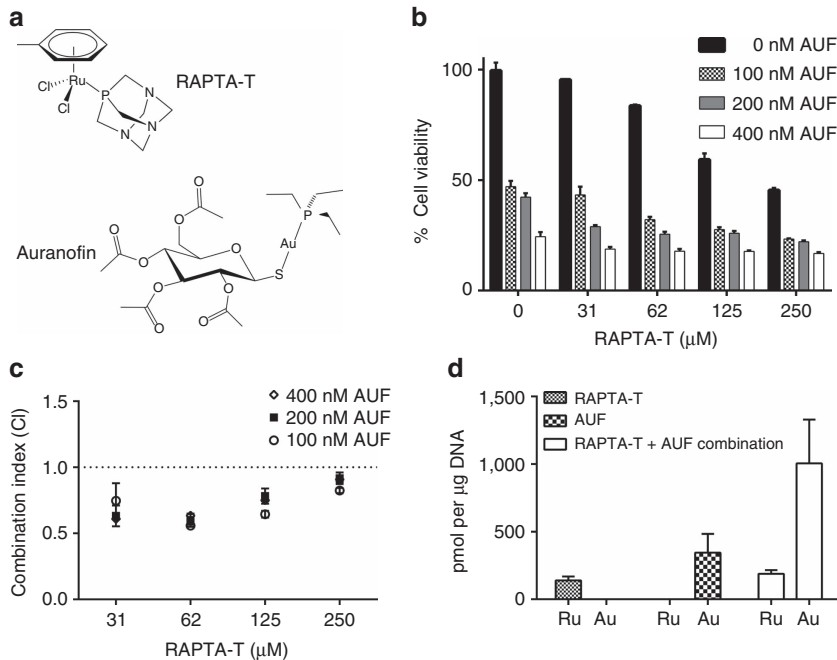

**Figure 1 | Synergistic activity of RAPTA-T and AUF in cancer cells.** (**a**) Structures of RAPTA-T and AUF. (**b**) Effect of combinations of RAPTA-T with AUF on cytotoxicity towards A2780 tumour cells (mean ± s.d., $n = 3$). (**c**) CI as a function of drug concentration (additive effect, CI = 0.9–1.1; slight synergism, CI = 0.7–0.9; synergism, CI = 0.3–0.7; strong synergism, CI = 0.1–0.3; mean ± s.d., $n = 3$). (**d**) Drug uptake into nucleosomes measured by ICP-MS after treatment with either RAPTA-T or AUF alone or the combination of the two (mean ± s.d., $n = 3$).

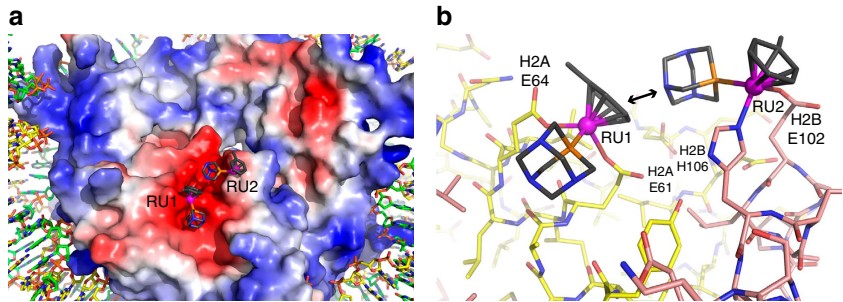

**Figure 2 | X-ray crystal structure of RAPTA-T–NCP.** (**a**) View of one face of the nucleosome core, with the histone octamer rendered with an electrostatic potential surface (red, negative; blue, positive). RAPTA-T binds within the extensive electronegative region on the H2A–H2B dimer, known as the acidic patch. (**b**) Structure of RAPTA-T-histone adducts. Ruthenium ion-coordinating side chains are labelled. H2A and H2B histone proteins are shown, respectively, with yellow and salmon coloured carbon backbones. An arrow indicates the van der Waals contact between the carrier ligands of the RU1 and RU2 adducts.

However, even from extensive treatments entailing 1 mM AUF in the buffer with an incubation time of up to 4 days, we found no evidence of gold adduct formation in the crystals (Supplementary Table 2). Since diffraction data were collected at the X-ray absorption edge of gold, sites of gold binding even at low occupancy can be detected.

We next conducted co-treatment trials with RAPTA-T and AUF simultaneously present in the treatment buffer and observed that both ruthenium and gold adducts are generated readily at distinct histone sites (Supplementary Tables 2 and 3; Supplementary Fig. 3). The RAPTA-T adducts form at the same two adjacent locations observed for the RAPTA-T only treatments, RU1 and RU2, whereas AUF generates adducts at two symmetry-related locations, AU1 and AU1′, which are not far from the central base pair and situated along the twofold axis of the nucleosome (Fig. 3a,b). The gold ions are coordinated to the imidazole delta nitrogen atoms of H113 and

H113′ from the two H3 histone proteins. The AUF sugar-thiolate groups have been substituted by the histone ligands, and the Au-triethylphosphine units fill a narrow, but shallow, access channel to the exposed nitrogen groups of the histidine side chains (Fig. 3c). The triethylphosphine groups make extensive hydrophobic contacts with surrounding H3 residues, which entail interactions with both copies of H3 for a given adduct.

**RAPTA-T adducts alter nucleosome conformational properties.** The requirement for the presence of both RAPTA-T and AUF in the NCP treatments to allow gold adduct formation at AU1 and AU1′ suggests that the RAPTA-T adducts promote reactivity of the H3/H3′ H113 sites. However, the closest distance between any two adduct atoms of RU1/RU2 and AU1/AU1′ is 27 Å (Fig. 3a; Supplementary Movie 1). Although no large conformational differences are observed between any of the three

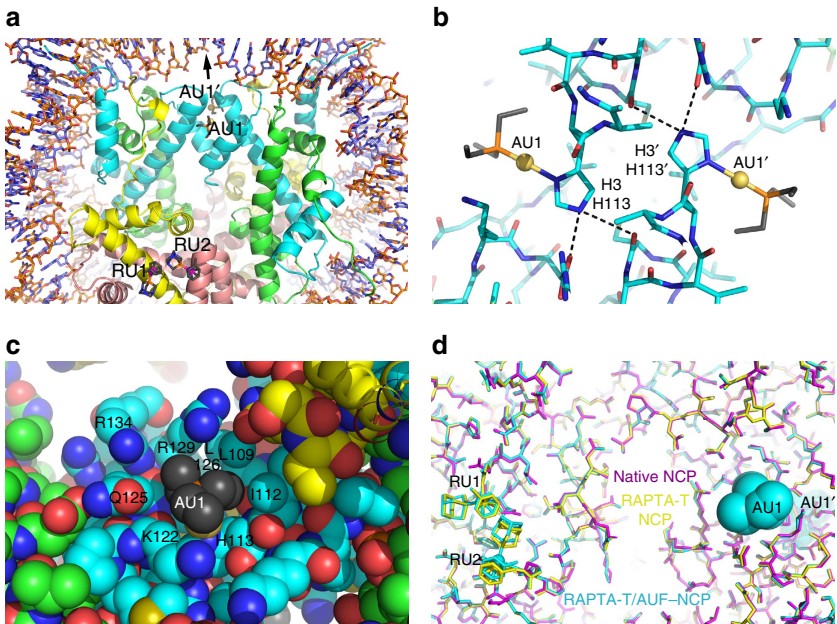

**Figure 3 | X-ray crystal structure of RAPTA-T/AUF–NCP.** (**a**–**c**) Structures of AUF and RAPTA-T adduct sites. H2A, H2B, H3 and H4 histone proteins are shown respectively with yellow, salmon, cyan and green backbone colouring. (**a**) Overview of AUF and RAPTA-T binding, illustrating the $\geq 27$ Å separation of the two types of adducts. The arrow indicates the pseudo-twofold symmetry axis of the nucleosome. (**b**) Structure of the AUF-histone adducts. Dashed lines indicate hydrogen bonding with the H3 H113 imidazole epsilon nitrogen groups. (**c**) The van der Waals environment of an AUF adduct, shown in space-filling representation. (**d**) Superposition of the native NCP (magenta), RAPTA-T–NCP (yellow) and RAPTA-T/AUF–NCP (cyan) models, illustrating that the structures of the RAPTA-T-containing models are nearly identical, whereas that of the native NCP differs subtly in the vicinity of where the adducts form.

crystallographic models—native NCP, RAPTA-T–NCP or RAPTA-T/AUF–NCP—there are small, but seemingly systematic, structural alterations associated with histone coordination of RAPTA-T (Fig. 3d). The perturbations coinciding with the RAPTA-T adducts appear to extend over some distance, including to within the immediate vicinity of the nearest AUF binding site (AU1).

To investigate whether there are systematic alterations in the structural properties of the nucleosome, which accompany RAPTA-T adduct formation and promote AUF reactivity, we conducted molecular dynamics (MD) simulations on four systems: the native NCP, RAPTA-T–NCP, RAPTA-T/AUF–NCP and AUF–NCP in aqueous solution. We observe that when the RAPTA-T adducts are present, the histone octamer undergoes a series of subtle conformational changes impacting both the RAPTA-T and AUF binding sites. MD simulations of the RAPTA-T–NCP and RAPTA-T/AUF–NCP complexes show that RAPTA-T induces a slight kink ($\sim 8^\circ$) within the long α-helix of H2A (Fig. 4a; Supplementary Fig. 4; Supplementary Movie 2). This perturbation of the local environment surrounding the RAPTA-T adducts occurs in the early steps of dynamics and is accompanied by further structural alterations. In fact, the H3 C-terminus (residues 133–135) that had not been resolved in the crystal structure due to its high mobility undergoes a conformational transition that culminates in multiple altered inter-histone contacts around the middle of the histone octamer (Fig. 4b; Supplementary Figs 5 and 6; Supplementary Movies 3 and 4). Furthermore, we observe systematic changes of the relative inter-helical orientations, which involve H2A and H3/H3′ α-helices and connect them via a network of subtle structural modifications that link the RU1/RU2 and the AU1/AU1′ adduct sites (Fig. 5). These specific RAPTA-mediated alterations in the conformation and dynamics of the histone octamer are seen for both the RAPTA-T–NCP and

RAPTA-T/AUF–NCP systems, but are neither observed for the native NCP nor the AUF–NCP systems.

As a consequence of the RAPTA-T adduct-mediated conformational changes, we observe a slight tightening of the structure at the level of the AUF sites. In fact, by analysing the nature of the hydrophobic interactions between the histone elements and the AU1/AU1′ adducts, we find that in the presence of RAPTA-T, AUF forms on average more contacts with the α-helices of both H3/H3′ histones than in the absence of RAPTA-T (Supplementary Fig. 7; Supplementary Tables 4 and 5). This indicates that when RAPTA-T adducts are present, histone substituents tend to make closer, more favourable hydrophobic contacts with the triethyl groups on the phosphine and suggests that the AUF binding sites are rendered on average more compact via the allosteric influence of RAPTA-T binding.

**Allosteric mechanism for promoting AUF adduct formation.** To detect the presence of possible dynamic correlations between the ruthenium and gold binding sites, as well as the molecular elements that could be responsible for 'signal transmission' between them, we performed a cross-correlation analysis. The cross-correlation matrix calculated for the histone components of the RAPTA-T/AUF–NCP system (Fig. 6a) shows a peculiar pattern of correlations, which is absent when RAPTA-T is not bound (Supplementary Figs 8, 9 and 10). These correlations describe a path among adjacent α-helices, which allows the transfer of information from RAPTA-T binding to the AUF sites. Starting with the kink of the H2A long α-helix (blue, panel i) that is induced by RAPTA-T binding, we find a sequence of high correlations among angles between adjacent α-helices, which connect the RU and AU sites (panels ii–vi). Indeed, the kink of the H2A long α-helix is highly correlated with the orientation of the H3′ long α-helix (magenta, panel ii;

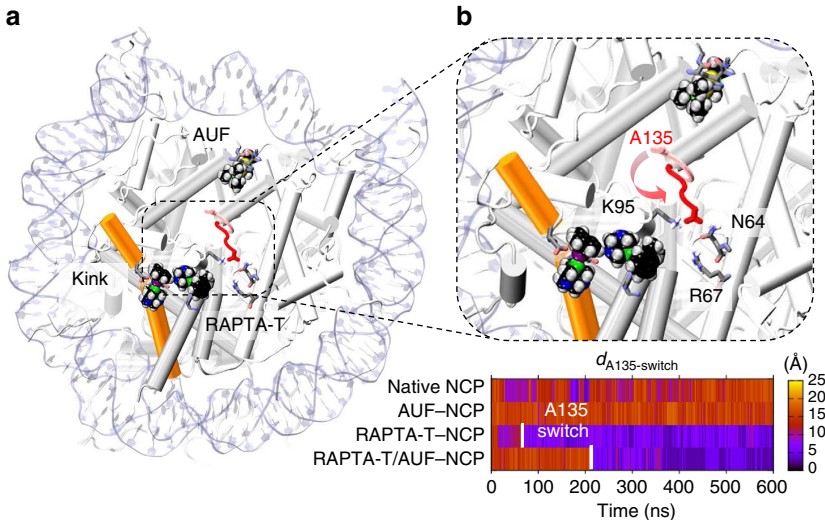

**Figure 4 | Conformational changes induced by RAPTA-T adducts.** (**a**) Snapshot from MD simulations of the RAPTA-T/AUF–NCP system, showing a kink within the long α-helix of H2A induced by the presence of RAPTA-T adducts. The histone proteins (grey) are shown as cartoon, highlighting the kinked H2A α-helix in orange. The DNA is represented as pale blue ribbons. RAPTA-T and AUF (black) are shown in space-filling representation. (**b**) Close view of the nucleosome core, showing the conformational change involving the H3 C-terminus (A135), which occurs in the presence of RAPTA-T adducts and culminates in the H-bonding of A135 with H2A K95 and H4 N64. An arrow indicates the conformational change of H3 A135, which is shown as sticks in its initial (pink) and final (red) configurations. The bottom graph reports the time evolution along MD of the simulated systems for the distance ($d_{A135\text{-switch}}$) between the Cα atom of H3 A135 and the centre of mass (COM) of the H4 N64 and H4 R67 residues, colour-coded according to the scale on the right. The A135 switch occurs after ∼80 ns (RAPTA-T/–NCP) and ∼220 ns (RAPTA-T/AUF–NCP) and is indicated with a white bar.

Pc = 0.86; Fig. 6b; Supplementary Fig. 11), which constitutes the AU1 binding site on the RAPTA-binding face of the structure. This latter element is further coupled to the α-helices of the H3-H3′ interface (panel iii), thus reaching the AU1′ site at the opposing face of the structure (Pc = 0.83). High couplings are also associated with the AU1-coordinating H3′ helix and elements of the H4 histone (panel iv) that couple to the central α-helices of H2A (panel v), which are in turn correlated with H2B elements (panel vi) and to the kink of the H2A α-helix that is induced by RAPTA-T binding (panel i).

Given the possibility that the interaction of the two RAPTA-T adducts with each other could alter the overall dynamics of the system, or that the presence of both RU1 and RU2 adducts are required for the observed effect, we also carried out an additional MD run of the RAPTA-T/AUF–NCP system with the RAPTA-T adduct at site RU2 omitted. Nevertheless, however, the ∼600 ns MD simulation shows that the single RAPTA-T adduct alone at site RU1 induces as well a kink in the long α-helix of H2A, like the one observed in the presence of both RU1 and RU2 RAPTA-T adducts (Supplementary Fig. 12). Furthermore, the A135 switch, which occurs after ∼220 ns in the simulation of the complete RAPTA-T/AUF–NCP system and after ∼80 ns for the full RAPTA-T–NCP system, is also observed after ∼90 ns in the RU1-only RAPTA-T/AUF–NCP system. Finally, the calculated cross-correlation matrix reveals a similar correlation pattern as observed in the simulation of the full RAPTA-T/AUF–NCP and RAPTA-T–NCP systems (Supplementary Figs 10 and 12). This indicates that the impact of RAPTA-T on altering the conformational dynamics in the nucleosome core can be achieved by just a single adduct residing in the acidic patch.

The cross-correlation evidence indicates the critical role of RAPTA-T in inducing a tight cooperation among the histone components, which culminates in the transmission of the allosteric effect to the AUF sites. Taken together, our correlation analyses indicate that, by inducing a kink in the H2A α-helix, RAPTA-T promotes the formation of highly correlated motions, thus allowing communication within the histone core and resulting in propagation of the effect of RAPTA-T binding to the AUF sites.

## Discussion

Crystallographic studies of proteins treated with different gold compounds, including triethylphosphine-Au-Cl, have shown that adducts form at specific histidine sites, in spite of the presence of thiol sulphur-donor groups[22,23]. In NCP crystals, there are at least five solvent accessible histidine imidazole groups available for metal compound binding[14–16], but we only observe AUF adducts at the one type of histidine site, H113 of H3/H3′ (and only when RAPTA-T is bound in the acidic patch). We have hitherto not observed metal compound binding at this site type, but these locations are unique in having a small access channel to the histidine delta nitrogen atom that is composed largely of hydrophobic histone elements (Fig. 3c). This apparently underlies the site selectivity of AUF in binding only to this particular motif, whereby association is fostered through hydrophobic histone interactions with the triethylphosphine group. Hydrophobic contacts, and more generally complementarity of shape between the metal ligands and protein substituents, appear to be major driving forces behind ruthenium/osmium compound-histone site selectivity, together with the chemical nature of the coordinating group(s)[14–17].

The enhancement of AUF reactivity towards nucleosome (chromatin) in the presence of RAPTA-T that we observe in the NCP crystals as well as in cancer cells is apparently also a consequence of modulating hydrophobic contacts and shape complementarity to favour binding, as evidenced by the MD simulations. The RAPTA-T binding results in dynamic remodelling of a major portion of the histone octamer in the nucleosome, which is linked to decreasing the average size of both of the H3/H3′ H113 binding pockets, yielding closer and more favourable hydrophobic contacts that promote AUF association. This allosteric mechanism behind the RAPTA-T adducts

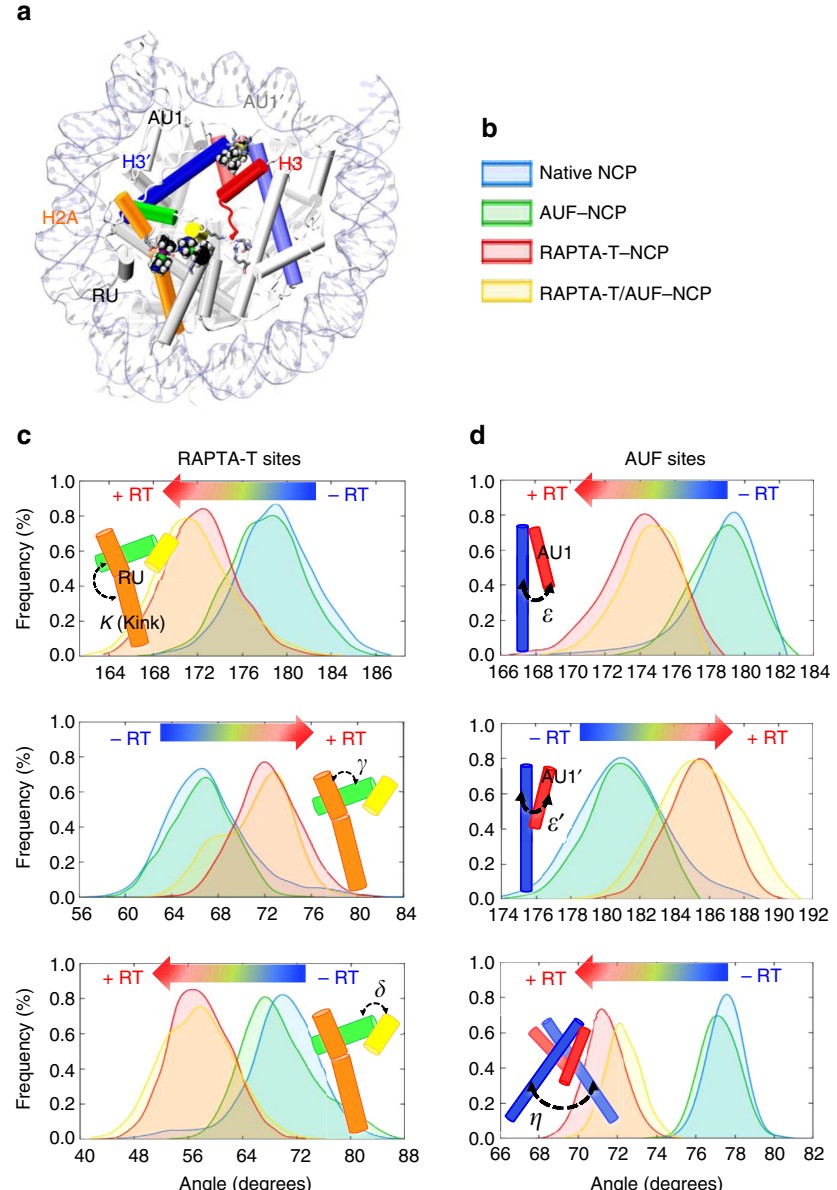

**Figure 5 | Histone α-helical rearrangements between the RAPTA-T and AUF adduct sites.** (**a**) Highlighting of dynamically coupled α-helices. (**b**) Colouring scheme for the different NCP systems shown in the probability distributions of panels **c** and **d**. (**c,d**) Probability distribution functions of the angles between the H2A (orange, green and yellow) and H3/H3′ (red, blue) α-helices that mediate an interface linking the RAPTA-T sites (**c**) and the AUF sites (**d**), calculated over the equilibrium trajectories of the native NCP (blue), AUF–NCP (green), RAPTA-T–NCP (red) and RAPTA-T/AUF–NCP (yellow) systems. A shift in the probability distributions is observed when RAPTA-T adducts are present (+RT) and is indicated with a blue-to-red arrow (−RT, adducts absent). A cartoon representation of the calculated inter-helix angles is shown within each graph, color-coded as given in the structure of the NCP in **a**.

promoting AUF adduct formation in the nucleosome provides a rationale for how co-treating cells with the two agents results in ~3-fold higher levels of gold adducts in chromatin relative to treatment with AUF alone.

Here we have characterized an example of a nucleosome-based drug–drug synergy as well as a small molecule-mediated allosteric effect in the nucleosome. There is consistency in our simulation results since we find effectively the same discrete dynamic transitions occurring in any of the three NCP systems containing RAPTA-T adducts (RAPTA-T–NCP, RAPTA-T/AUF–NCP and RAPTA-T/AUF–NCP with only a single RAPTA-T adduct at the RU1 site), and we do not observe these conformational features in either of the systems lacking RAPTA-T adducts (native NCP and AUF–NCP). In fact, a very recent MD simulation-based study showed that there are allosteric networks within the nucleosome core, which can be modulated by histone variant changes[24]. Specifically, it was found that substitution of just four amino acid residues in the histone-fold loop 1 (L1) region play a dominant role in yielding differential dynamics and energetics between nucleosomes composed of the ubiquitous H2A versus the H2A.Z and macroH2A variants. Moreover, sites of histone posttranslational modification are disproportionately prevalent at key locations within the allosteric networks. In conjunction with our findings here, this indicates that there are likely to be multiple allosteric mechanisms within the nucleosome, which have the potential to modulate

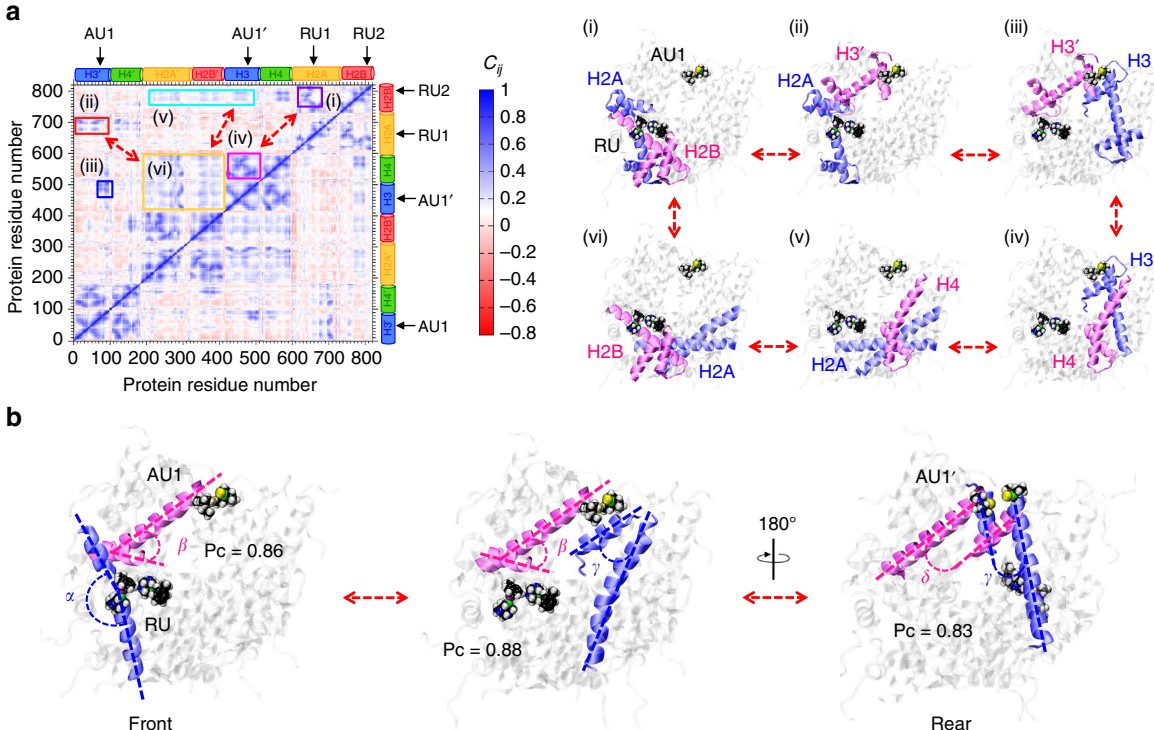

**Figure 6 | Allosteric mechanism mediating cross-talk between RAPTA-T and AUF sites.** (**a**) Cross-correlation matrix of the fluctuations of the Cα atoms ($C_{ij}$) around their mean positions, calculated over the equilibrium MD trajectory of the RAPTA-T/AUF–NCP system. The extent of correlated ($0 > C_{ij} < 1$) and anticorrelated ($-1 > C_{ij} < 0$) motions is color-coded according to the scale on the right. RAPTA-T (RU1/RU2) and AUF (AU1/AU1′) sites are indicated, as well as the H2A, H2B, H3 and H4 histones. Highly correlated regions are highlighted within the panels (i–vi). Histone protein components involved in the (i–vi) correlations are shown on the right. Histones are shown in cartoon representation, with correlated residues highlighted in blue and magenta. RAPTA-T and AUF are in space-filling representation. (**b**) Graphical representation of Pearson correlation coefficient (Pc) analysis used to compute the strength of coupling between the dynamics of the α-helices of the histones for the RAPTA-T/AUF–NCP system. The highest Pc calculated between pairs of angles formed by the adjacent histone α-helices are reported, revealing a correlation path that connects the H2A α-helix kink, occurring at the RU sites, with the AUF sites at both the front (AU1) and rear (AU1′) of the nucleosome. Blue and magenta dashed lines are used to indicate the correlated angles. The structure is rotated along the pseudo-twofold axis, showing the front face (left and middle) and rear face (right) of the NCP.

chromatin activity and can be influenced by even subtle chemical changes to the histone proteins.

The allosteric mechanism at play could be dependent on the specific nucleosome binding event or alteration in question, but may involve one of either two proposed classifications[7]. In the 'domino' model, a sequential set of local events propagates via a well-defined pathway from one allosteric active site to the other, spatially distant, site through a network of highly correlated neighbours. In contrast, with the 'violin' model, the binding event alters a collective vibrational mode of the molecule, transferring the signal through the whole system without a specific route. The 'signal transmission' we observed here from the ruthenium to the gold binding sites in the nucleosome core appears to occur mainly via a domino form of events. Although signal transmission does not seem to be limited to a single pathway, it is possible to identify just a few pathways with particularly high local correlations, in which the binding of RAPTA-T at the RU1 site induces a local perturbation (kink in the long α-helix of H2A), creating a defined sequence of correlations among angles between adjacent α-helices, thereby connecting the RU and AU sites.

Interestingly, comparison of the stoichiometric gold versus ruthenium content in chromatin for the RAPTA-T/AUF co-treatments shows that there is more than a fivefold greater quantity of gold. In the RAPTA-T/AUF–NCP crystal structure, there are two equivalents each of ruthenium and gold per nucleosome. RAPTA-T adducts form on only one face of the

nucleosome, meaning at a single H2A–H2B dimer, as a result of the symmetry-related (RU1′/RU2′) sites on the other H2A–H2B dimer being sterically blocked due to crystal contacts[14]. Nonetheless, the MD studies suggest that the allosteric impact of a RAPTA-T adduct on only one face of the nucleosome is sufficient to influence both of the AUF binding sites in a similar fashion. As such, *in vivo*, it is possible that RAPTA-T adducted and non-adducted H2A–H2B dimers are exchanged between different nucleosomes[25], which thereby effectively facilitates AUF adduct formation across chromatin. This mechanism could explain the excess of AUF equivalents we observe in the cellular chromatin. Beyond this, adduct occupancy levels on a given nucleosome in the cell may differ with respect to the crystallographic studies, where high compound concentrations are used to ensure acquisition of accurate structural models. Nevertheless, we have observed that RAPTA compounds and certain other ruthenium-based agents form adducts first at the RU1 site[14,16], and moreover occupation of this site alone appears sufficient to elicit the allosteric effect. This suggests that a single RAPTA-T adduct per nucleosome would be sufficient to mediate the allosteric activity *in vivo*. One additional possibility behind the disproportionately high gold versus ruthenium levels observed on the isolated cellular chromatin could relate to AUF adducts forming at sites outside of the nucleosome core, including for instance binding to linker histones, which we cannot presently rule out.

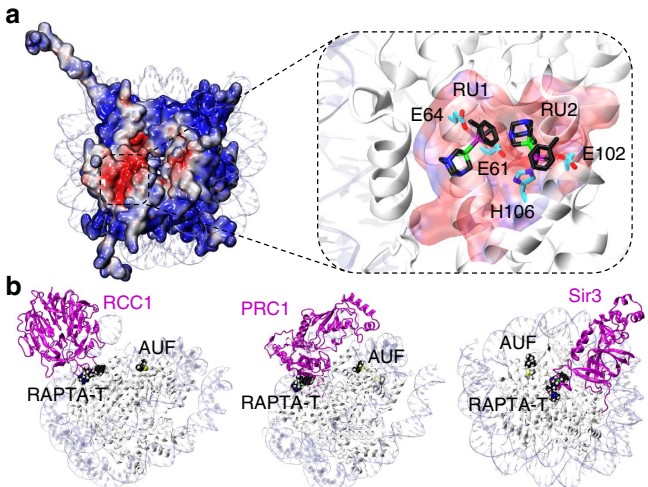

**Figure 7 | Potential for RAPTA-T and AUF adducts to interfere with nucleosome-nuclear factor interactions.** (**a**) Electrostatic potential (red, negative ($-5\,kTe^{-1}$); blue, positive ($+5\,kTe^{-1}$)) of the histone protein octamer in the native NCP[38], highlighting the H2A–H2B acidic patch (boxed region). A close-up view of the acidic patch is shown in the inset, emphasizing the central location of the RAPTA-T adducts at sites RU1 and RU2 in the RAPTA-T/AUF–NCP crystal structure. RAPTA-T (black C backbone) and coordinating protein residues (cyan C backbone) are shown as sticks. (**b**) NCP-on-NCP superpositions of the RAPTA-T/AUF–NCP crystal structure with those of either the regulator of chromatin condensation 1 (RCC1)[26], polycomb repressive complex 1 (PRC1)[27] or silent information regulator 3 (Sir3)[57] chromatin proteins bound to the NCP (the NCP and the second chromatin factor molecule associated with the other nucleosome face in the assembly are omitted for clarity). Chromatin factors (magenta) are shown as ribbons and RAPTA-T and AUF in space-filling representation.

The synergistic effect of RAPTA-T and AUF in mediating disproportionately high levels of AUF adducts in cellular chromatin is consistent with the nucleosome allosteric mechanism we have determined to be in operation *in vitro*. The synergistic cytotoxic effect of RAPTA-T/AUF to cancer cells may also be linked to the higher chromatin adduct levels. It is likely that the synergy in chromatin adduct formation is at least partially responsible for the cytotoxicity synergism, since we observe that pre-treatment of cells with RAPTA-T has a sensitizing effect on the tumour cells, and it is the RAPTA-T-chromatin adducts that would have to be initially present to promote AUF adduct accumulation. Taken together in conjunction with our previous related studies, this strongly suggests that adducts on the histone proteins can have a therapeutic impact[14–18], such as the increased tumour cell cytotoxicity observed here. As we had previously proposed, adducts within the acidic patch have the potential to interfere directly with nuclear factor binding and chromatin dynamics and thereby mediate a cytotoxic effect. In particular, the glutamate residues involved in coordinating RAPTA-T at the RU1 site are part of a seemingly ubiquitously employed nucleosome-recognition motif that binds one or more basic residues of nuclear proteins that interact with the acidic patch[19–21]. Therefore, adducts that form here could inhibit or block the binding of chromatin associating factors, such as the regulator of chromatin condensation 1 (ref. 26) and polycomb repressive complex 1 (ref. 27) proteins (Fig. 7; Supplementary Fig. 13). Although AUF reacts at sites well outside of the acidic patch, an adduct at either one or both AU1/AU1′ sites also alters the surface properties of the histone octamer in such a way that could influence factor

binding. In fact, a model based on the structure of the polycomb repressive complex 1 ubiquitylation module bound to the NCP places the ubiquitin substrate[27] in close proximity with the AU1/AU1′ sites (Supplementary Fig. 14). This suggests that the AUF adducts could also interfere with nuclear factor transactions and in conjunction with activities mediated by the RAPTA-T adduct(s) may underlie at least in part the synergistic cytotoxic effect we find for the two drugs.

The allosteric mechanism mediating drug–drug synergy we have characterized in this work suggests that there is untapped potential for therapeutic modulation of chromatin activity through exploitation of structural and dynamical features of the nucleosome. Although we have made the initial discovery here with metal-based drugs, which are especially favourable for structural visualization and cellular tracking, it is likely that synergies and allosteric actions in the nucleosome occur with other drug classes as well. Moreover, our findings also suggest that the site targeting and associated activities of drugs administered individually can differ from those corresponding to the drug combination. In a broader sense, the structural and dynamics impact of drug binding to the nucleosome acidic patch, a chromatin protein binding hot-spot[19–21], implies that analogous histone-mediated allosteric phenomena[24] may be in operation for genomic regulation *in vivo*. This yields new possibilities for the potential of nuclear factor binding or histone posttranslational modifications to alter nucleosome activity over a distance.

## Methods

**Metal compounds.** The synthesis of RAPTA-T was carried out as previously described[10]. Briefly, equimolar amounts of 1,3,5-triaza-7-phosphatricyclo [3.3.1.1]decane and [Ru($\eta^6$-$p$-toluene)Cl$_2$]$_2$ were refluxed in methanol, and the product was precipitated by addition of diethyl ether. The purity of the prepared product was confirmed through elemental analysis. AUF was purchased from Alexis Biochemicals (Switzerland).

**Crystallographic analysis of treated nucleosome core particle.** X-ray crystallographic analysis of NCP was conducted using nucleosome assembled with recombinant histones and a 145 bp DNA fragment[28]. NCP crystals were grown by the hanging droplet method using buffers of MnCl$_2$, KCl and K-cacodylate (pH 6.0), and crystals were subsequently stabilized in harvest buffer (37 mM MnCl$_2$, 40 mM KCl, 20 mM K-cacodylate (pH 6.0), 24% 2-methyl-2,4-pentanediol and 2% trehalose)[29]. The 37 mM MnCl$_2$ buffer component was subsequently eliminated by gradual replacement with 10 mM MgSO$_4$ followed by thorough rinsing of crystals with the MgSO$_4$-containing buffer to remove any residual MnCl$_2$ (ref. 14).

The RAPTA-T–NCP crystal structure described in the main text stems from a 20-h incubation of NCP crystals with 0.25 mM RAPTA-T included in the buffer (Supplementary Table 1; data collected at an X-ray wavelength of 1.50 Å). The RAPTA-T/AUF–NCP crystal structure described in the main text stems from a 40-h incubation of NCP crystals with 2 mM AUF and 0.5–1 mM (22 h at 0.5 mM and 18 h at 1 mM) RAPTA-T included in the buffer (Supplementary Table 3; data collected at an X-ray wavelength of 1.50 Å). Single-crystal X-ray diffraction data were recorded, subsequent to mounting stabilized crystals directly into the cryocooling N$_2$ gas stream set at $-175\,^\circ$C (ref. 16), at beam line X06DA of the Swiss Light Source (Paul Scherrer Institute, Villigen, Switzerland) using a Pilatus detector and X-ray wavelengths of either 1.04 or 1.50 Å. Data were processed with MOSFLM (ref. 30) and SCALA from the CCP4 package[31].

The 2.5 Å resolution crystal structure of NCP containing RAPTA-C (C = cymene) adducts (*pdb* code 3MNN)[14] was used for initial structure solution by molecular replacement. Structural refinement and model building were carried out with routines from the CCP4 suite[31]. Restraint parameters for the adducts were based on the small molecule crystal structures of AUF (ref. 32) and RAPTA (refs 33,34) compounds. Data collection and structure refinement statistics are given in Supplementary Tables 1 and 3. Graphic figures were prepared with PyMOL (DeLano Scientific LLC, San Carlos, CA, USA).

**Cell culture maintenance.** Human ovarian A2780 (ECCAC, Salisbury, UK) cancer cells were cultured in Roswell park memorial institute (RPMI)-1640 medium (Life Technologies, Switzerland) supplemented with 10% foetal bovine serum (FBS, Sysmex, Horgen, Switzerland). Cells were maintained in a humidified environment (37 °C, 5% CO$_2$).

**Cellular viability assay.** Cell viability was determined using the Presto Blue assay. Cells were seeded in 96-well plates as monolayers with 100 μl of cell suspension (~5,000 cells) per well and preincubated for 24 h in medium supplemented with 10% FBS 37 °C and 5% CO₂ 100 μl of the compound solutions in RPMI medium containing 10% FBS or RPMI medium containing 10% FBS (for controls) were added to each well, and the plates were incubated for 72 h. In the case of combination assays using pre-treatment conditions, the plates were incubated for 3 h, then the supernatant was aspirated and replaced by 100 μl of the respective other compound solutions and incubated for another 48 h.

Subsequently, Presto Blue was added to the cells (10 μl per well) and the plates were incubated for 1 h. The fluorescence intensity, directly proportional to the number of surviving cells, was quantified at ex560 nm/em590 nm using a multiwell plate reader (Molecular Devices), and the fraction of surviving cells was calculated from the fluorescence intensity of untreated control cells. Evaluation is based on the mean from at least two independent experiments, each comprising triplicates per concentration level.

**Analysis of cellular cytotoxicity synergy.** Drug-induced cytotoxicity synergy was analysed with the CompuSyn software (Biosoft, Cambridge, UK), in which the synergy is expressed as a combination index (CI). The CI method for quantifying drug cytotoxicity synergism is based on the approach of mass-action law and the median-effect principle derived from enzyme kinetic models developed by Chou and Talalay, which has been used extensively to evaluate drug interactions[35–37]. In the application of the CI approach, synergism is defined as a combined effect that is statistically significantly greater than the purely additive effect of the individual components, whereas antagonism is defined as a combined effect that is statistically significantly less than the purely additive effect of the individual components. As such, in the application of this approach, a CI = 1 indicates a purely additive effect, a CI < 1 indicates synergy and a CI > 1 indicates antagonism. The creators of the CompuSyn software have proposed that CI values be interpreted as follows: CI > 1.1, antagonistic effect; CI = 0.9–1.1, purely additive effect; CI = 0.7–0.9, mild synergism; CI = 0.3–0.7, moderate synergism; CI = 0.1–0.3, strong synergism; and CI < 0.1, very strong synergism.

**Isolation of cellular chromatin.** Cells were grown at 37 °C and 5% CO₂ in RPMI 1640 medium containing 10% FBS and 1% penicillin/streptomycin, 150 cm² flasks to ~80% confluency. Then, cells were incubated with RAPTA-T at a concentration of 500 μM or 20 μM auranofin or the combination of both at 37 °C for 2 h. Subsequently, the cells were washed with ice-cold phosphate-buffered saline solution to remove unbound drug and chromatin was extracted using a Pierce Chromatin Prep Module (Thermo Fisher Scientific, Switzerland) according to the manufacturer's protocol. Three independent experiments were performed for each treatment.

**Quantification of cellular DNA.** All samples were analysed for their chromosmal DNA content before ICP-MS measurements. DNA was quantified by the PicoGreen dsDNA quantitation assay (Invitrogen). PicoGreen (50 μl per well, 200× diluted in 10 mM Tris + 1 mM EDTA buffer) was added to 50 μl of DNA sample and the fluorescence signal was determined by spectrofluorometric analysis (484 nm excitation/520 nm emission) using an automated reader (SpectraMax5e, Molecular Devices).

**Inductively coupled plasma mass spectrometry measurements.** Au, Ru and In standard solutions (1 g l⁻¹ in 2% HCl, 2% HNO₃ and 10% HCl, respectively) were purchased from CPI International (Amsterdam, The Netherlands). Hydrochloric acid (37%) at high-purity grade was purchased from Merck (Darmstadt, Germany).

Before determination of ruthenium and gold content, the chromatin extracts were digested with 400 μl of 37% hydrochloric acid solution overnight at room temperature and adjusted with ultrapure water to a final volume of 4 ml. Indium was added as an internal standard at a concentration of 0.5 ppb. Determinations of total metal contents were achieved on an Elan DRC II ICP-MS instrument (Perkin Elmer, Switzerland) equipped with a Meinhard nebulizer and a cyclonic spray chamber. The ICP-MS instrument was tuned daily using a solution provided by the manufacturer containing 1 ppb each of Mg, In, Ce, Ba, Pb and U. External standards were prepared gravimetrically in an identical matrix to the samples (with regard to internal standard and hydrochloric acid) with single element standards.

**Structural models for computational investigations.** Five simulation systems of the NCP were built: the native form (NCP), with two AUF adducts (AUF–NCP), with two RAPTA-T adducts (RAPTA-T–NCP), with two AUF and two RAPTA-T adducts (RAPTA-T/AUF–NCP) and with two AUF and a single RAPTA-T adduct (RAPTA-T/AUF–NCP, RU1 site only; RU2 adduct omitted). These simulation systems were based on the crystal structure of the native NCP (PDB code 1AOI, solved at 2.80 Å resolution)[38] and the crystal structures of RAPTA-T–NCP and RAPTA-T/AUF–NCP reported here. Missing residues of the histone proteins in the RAPTA-T–NCP and RAPTA-T/AUF–NCP structures—consisting of Ala135 (C-terminus of H3), Lys16–Lys20 (H4), Gly4–Ala14, Lys119 (H2A) and

Lys24–Lys28 (H2B)—were added in conformity with the crystal structure of the native NCP (ref. 38). Residues from Tyr120 to the C-terminus of the H2A histone (Lys127) were modelled from the primary sequence using Modeller 9.13 (ref. 39). Each model system was embedded in explicit waters, while Na⁺ counter-ions were added to neutralize the total charge, leading to a periodic box of ~160 × 120 × 150 Å³ and a total number of ~313,000 atoms for each system.

**Classical molecular dynamics simulations.** Classical MD was used to equilibrate the aforementioned systems [(1) NCP, (2) AUF–NCP, (3) RAPTA-T–NCP, (4) RAPTA-T/AUF–NCP and (5) RAPTA-T/AUF–NCP, RU1-only] under physiological conditions and for the production runs. The AMBER force field ff99SB2 (ref. 40) with the ff99Bildn3 (ref. 41) modifications was used for the histone proteins, whereas the parmbsc0 (ref. 42) modification of the AMBER parm99 force field was adopted for the DNA. The TIP3Pn (ref. 43) model was employed for the description of explicit waters. Non-standard parameters for RAPTA-T were taken from previous studies[44]. The AUF residue was treated with the general Amber force field[45]. The bond and angle parameters for the N–Au–P fragment were obtained from vibrational frequencies after geometry optimization of the complex at the B3LYP/6-31 + G* level, using the relativistic effective core LANL2DZ pseudopotential for gold. Van der Waals parameters for Au(I) were taken from Allinger et al.[46]. The atomic partial charges were derived following the Merz-Kollman scheme (RESP)[47]. During the RESP fitting procedure, the charges of the Ru and Au metal centres and of their coordinating ligands were fixed to the values obtained from a Bader charge analysis[48].

The systems were simulated with a time step of 1–1.5 fs, as detailed below in the description of the simulations protocol. All simulations were performed using the Gromacs 4.6.3 code[49]. The LINCS algorithm[50] was used to constrain covalent bonds involving hydrogen atoms. Long-range electrostatic interactions were calculated with the particle mesh Ewald method with a real space cut-off of 10 Å. Periodic boundary conditions in the three directions of Cartesian space were applied. The systems were coupled to a Nosé − Hoover thermostat[51,52] at a reference temperature of 310 K and to an isotropic Parrinello − Rahman barostat[53] at a reference pressure of 1 bar, both with coupling time constants of 1 ps. The following simulation protocol was adopted for each of the four systems. First, the systems were subjected to energy minimization using a steepest descent algorithm. Then, the systems were thermalized to the physiological temperature of 310 K within 1,000 ps. During this phase, a time step of 1 fs was employed. Then, ~8 ns of MD were carried out using a time step of 1 fs, while subsequent runs were performed using a 1.5 fs time step, following a protocol that we had employed in our previous studies on ruthenium anticancer agents binding to the NCP (refs 16,54), as well as to naked DNA[44,54].

Approximately 700 ns of MD simulations were collected in the isothermal-isobaric (NPT) ensemble under standard conditions, for each of the first four systems, resulting in a total of ~2.8 μs of dynamics. In detail, ~750 ns of MD were carried out for the RAPTA-T/AUF–NCP and RAPTA-T–NCP systems; while ~700 ns of MD were conducted for the RAPTA-T/AUF and the native NCP systems. Then, the first ~100 ns were removed and analysis was performed on the subsequent ~600 ns of MD, for each simulated system. Finally, to study the effect of the binding of just a single RAPTA-T molecule, an additional MD simulation of the RAPTA-T/AUF–NCP system was performed including only a single RAPTA-T molecule at site RU1. This system was also simulated for a total of ~700 ns, removing the first ~100 ns of MD from the analysis. Coordinates of the systems were collected every 10 ps, for a total of ~70,000 frames for each run. All structural analyses were performed on the equilibrated trajectories.

**Probability distribution analysis.** The probability density functions of the angles between the α-helices, reported in Fig. 5 and Supplementary Fig. 12, were estimated by using a kernel density estimation. Kernel density estimation is a non-parametric way to estimate the probability density function of a random variable. A kernel is a smooth function centred at the location of each data point. The contributions from each function are summed and plotted. Mathematically, the kernel estimate $f(x)$ for a data set of $N$ points $x_i$ is:

$$f(x) = (1/Nh)\Sigma_i^N K((x - x_i)/h) \tag{1}$$

where $K(x)$ is any smooth, normalized function, and $h$ is the bandwidth: a measure of the width of the kernel function. Importantly, the angles between α-helices of the histone components were calculated considering as α-helical axis the principal axis of inertia of the Cα atoms of the amino acids forming the helix. Probability density functions were calculated over the production runs (that is, 600 ns of MD) for all simulated systems.

**Cross-correlation analysis.** The cross-correlation matrix $C_{ij}$ between the fluctuations of the Cα atoms relative to their average positions was used to identify the coupling of the motions between the protein residues. $C_{ij}$ was calculated from the last ~200 ns of MD simulations of each of the simulated systems [(1) NCP, (2) AUF–NCP, (3) RAPTA-T–NCP, (4) RAPTA-T/AUF–NCP and (5) RAPTA-T/AUF–NCP, RU1-only], using equation (2), where $\Delta\mathbf{r}_i$ and $\Delta\mathbf{r}_j$ are the fluctuation vectors of the atoms $i$ and $j$, respectively. The angle

bracket represents an average over the sampled period. The value of $C_{ij}$ ranges from $-1$ to 1. Positive $C_{ij}$ values represented a correlated motion between atoms $i$ and $j$, while negative $C_{ij}$ values describe anticorrelated motions. Correlations were calculated from an MD trajectory using the programme Carma 1.4 (ref. 55).

$$C_{ij} = \frac{\langle \Delta \mathbf{r}_i(t) \cdot \Delta \mathbf{r}_j(t) \rangle}{\left( \langle \Delta \mathbf{r}_i(t)^2 \rangle \langle \Delta \mathbf{r}_j(t)^2 \rangle \right)^{1/2}} \quad (2)$$

The cross-correlation matrices were calculated from the last $\sim 200$ ns of MD for each system to allow for a consistent comparison among the four different MD runs (Supplementary Fig. 8). To address the issue of reproducibility and convergence within the $\sim 200$ ns time windows reported in Supplementary Fig. 8, we also calculated the $C_{ij}$ matrix in time windows of $\sim 400$ ns, and over the entire production runs (shown in Supplementary Fig. 9). Specifically, the time windows considered refer to: (i) the last $\sim 200$ ns of MD; (ii) the last $\sim 400$ ns of MD and (iii) the entire production run, comprising $\sim 600$ ns of MD. Subsequently, the mean square deviations of the matrix elements (that is, $\Delta(C_{ij})^2$), as well as the associated root mean square deviations, were computed, following a testing procedure that we have employed for other MD simulations on protein/nucleic acids systems[56]. As a result, the computed root mean square deviation values range from $\sim 0.012$ to 0.019, indicating that the MD results are robust and reproducible within the time frames considered for analysis (Supplementary Fig. 9).

**Pearson correlation coefficient analysis.** A Pearson correlation coefficient (Pc) analysis was used to compute the strength of coupling between the α-helices of the histone components of the different systems studied. Pc are defined as in equation (3), where $\overline{X}$ and $\overline{Y}$ are averages of the time series.

$$\text{Pc} = \frac{\sum_i^n (X_i - \overline{X})(Y_i - \overline{Y})}{\sqrt{\sum_i^n (X_i - \overline{X})^2} \sqrt{\sum_i^n (Y_i - \overline{Y})^2}} \quad (3)$$

Pc were calculated between all pairs of angles formed by the histone α-helices and sorted with respect to increasing correlations Pc ($0.6 < \text{Pc} < 1$).

Pc values were calculated between all pairs of angles formed by the histone α-helices. Then, by considering the highest Pc ($0.6 < \text{Pc} < 1$), we detected two possible pathways that allow the transfer of binding information from the RAPTA-T (RU) sites to the AUF (AU1/AU1′) sites (Supplementary Fig. 11). Although the highest Pc values track a preferred pathway (Path A, shown in Fig. 6b), we located an alternative pathway (Path B), which also connects the kink of the H2A histone with the front and rear of the nucleosome core structure.

**Data availability.** Atomic coordinates and structure factors for the RAPTA-T–NCP and RAPTA-T/AUF–NCP models are deposited in the Protein Data Bank under accession codes 5DNM and 5DNN, respectively. Other data that support the findings of this study are available from the corresponding authors upon request.

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

## Acknowledgements

We thank M. Wang, V. Olieric and staff at the Swiss Light Source (Paul Scherrer Institute, Villigen, Switzerland). We are grateful for financial support by the Singapore Ministry of Health National Medical Research Council (Grant NMRC/1312/2011) and Ministry of Education Academic Research Fund Tier 3 Programme (Grant MOE2012-T3-1-001). This research was supported by the Swiss National Science Foundation via individual Grant No. 200020-140865 and the NCCR Chemical Biology.

## Author contributions

Z.A. and Z.M. carried out the NCP X-ray crystallographic work; G.P. conducted the computational investigations; T.R. carried out the cellular studies; R.M. contributed to the RAPTA-T–NCP crystallographic work; U.R. designed and supervised the computational part; P.J.D. designed the research and supervised the cellular studies and the overall project; C.A.D. supervised the crystallographic studies and conducted manuscript writing; All authors provided critical commentary and contributed to editing the manuscript.

## Additional information

**Competing interests:** The authors declare no competing financial interests.

