## [Peer Review File · Nature Communications]

Reviewers' comments:

Reviewer #1 (Remarks to the Author):

In this paper, Adhireksan et al. examine the synergistic effects of two drugs, RAPTA-T and AUF, on cancer cells and chromatin fibers. They use multiple methods, including cell assays, crystallographic studies, and molecular dynamics simulations, to make the case that RAPTA-T causes a conformational change in the NCP that promotes AUF binding. Overall this is a very nice piece of work, it integrates many different methods together and helps advance the emerging paradigm that the NCP core is dynamic in nature with allosteric networks that may be important for epigenetic regulation. This paper will be of interest to a wide range of experimental and theoretical groups working on chromatin biology, and has the potential to be very high impact. The claims are largely novel, although as discussed below they are not placed in the proper context of the scientific literature and do build on related studies that are not cited. For example, the phenomenon of allostery in the NCP core was recently discussed and analyzed in a biophysical journal paper, which is discussed below. In addition, while the claims are mostly convincing, there are areas that need significantly better justification, such as the choices made in setting up the MD simulations. Given the promise of the work in this manuscript, I recommend allowing the authors the opportunity to revise their manuscript and resubmit to Nature Communications.

My two most major concerns are:

First, the authors have taken the binding states from the crystallographic studies and used them as starting states for their MD simulations. This makes the (major) assumption that the binding states observed in the crystal structures are the biologically relevant states. Given that the crystal structures have multiple ligands bound to the NCP, and that the crystallographic conditions have significantly higher drug concentrations than the in vitro experiments, there is a strong possibility that the crystal structures have more drug molecules bound to the NCP than is biologically relevant. Therefore, it's not clear that the effects observed in the MD simulations are representative of the in vivo NCP dynamics. This is especially worrisome for the RAPTA-T molecules, as the simulations have two drug molecules in very close proximity to one another, but if only one is bound at physiologically relevant concentrations then the NCP dynamics are likely to be significantly different. Also, for AUF the experimental IC₅₀ is four orders of magnitude lower than the concentration used in the crystallographic medium, again suggesting that in vivo only one AUF may be bound to the NCP and not the two observed in the crystal structure. The authors never address these possibilities. This manuscript needs to include evidence and a strong justification that the most biologically relevant state includes four ligand molecules, with two RAPTA-T molecules on one face of the NCP and one AUF molecule on each face.

Second, The overall results are very exciting, however the authors should make an effort to examine their work in the wider scope of the field. For example, many models for allosteric regulation have been developed; how does the mechanism they observe fit in with these models? In addition, the phenomenon of histone-mediated allosteric pathways as a mechanism of epigenetic regulation has recently been proposed and extensively examined in MD simulations in the following paper:

"Effects of MacroH2A and H2A. Z on Nucleosome Dynamics as Elucidated by Molecular Dynamics Simulations," S Bowerman, J Wereszczynski, Biophysical journal 110, 327

The authors should discuss how their study fits in with the results of this manuscript.

Other, more minor issues include:

The authors used ICP-MS to determine that RAPTA-T and AUF accumulate on chromatin fibers, and they then make the jump to saying this means they bind to the NCP. It's possible there are higher

affinity binding locations for the drug molecules at other sites in chromatin, such as in linker DNA, linker histones, or other chromatin associated proteins, and that at biologically relevant concentrations significant fraction of these drugs bind to these alternative sites. This is especially problematic for AUF molecules, for which the only evidence of direct NCP binding is the crystal structure that was solved at AUF concentrations significantly higher than the IC50 for this molecule. The authors should discuss these possibilities.

The methods section contains a large amount of important information and should be moved to the main text.

From the methods section it is unclear what the exact protocol was for the MD simulations. Which simulations had a 1 fs timestep, and which had a 1.5 fs timestep? Exactly how long were the simulations, how much was removed for equilibration, and how much was used for each analysis method? Is the equilibration time removed from the plots in Figures 4 and S5? Giving approximate times isn't sufficient for these details.

The authors should make a convincing case that their results are reproducible and not the benevolent result of a single simulation trajectory.

The PDBs 5DNM and 5DNN should be made available to reviewers.

Why were only the last 200 ns of the simulations used for the cross correlation analysis? How reproducible are these analyses on other portions of their trajectories?

It would be helpful if the authors provided tables of the cross-correlation values between the key NCP locations and drug molecules. How strong are the correlations between the four drug molecules? These are difficult to read from the plots shown in Figure S8.

Figure S7 is difficult to interpret without error bars. Are these results statistically significant?

The authors should provide some discussion about possible mechanisms of action for these drugs. Why would binding at these sites on the NCP result in cell death?

Reviewer #2 (Remarks to the Author):

A. Results presented here discuss the potential for drug-drug synergism within chromatin and show actual drug binding within a single NCP. The two drugs (RAPTA-T and AUF) seemingly bind a independent sites, though there are order of addition effects that are seen where RAPTA-T binding seems to have a positive effect on the binding of AUF. Also, RAPTA-T does not display very useful cytotoxicity on its own in A2780 cells, but in combination with AUF, RAPTA-T facilitates killing of cells. This effect is most noticeable below 400 nM AUF, where the effect seems to drop off.

B. The work's true novelty lies with the notion that measurable drug-drug synergy within chromatin has not been previously reported, though it has in certain signaling pathways. The synergy reported here, is not that of distant molecules in a pathway, but neighboring sites on a single NCP. The real synergy is that binding of the 1st molecule helps the 2nd molecule bind in larger amounts. How this exactly leads to greater killing of cancer cells isn't as clear.

C. Data and presentation are clear and reported in a understandable fashion.

D. All stats seem appropriate.

E. The conclusions reported generally follow the data presented. The authors do a nice job of

attempting to show a connection between the two drug binding sites using crystallography, MD, and correlation stats. The authors spend much effort in convincing the reader that binding of RAPTA-T sends a signal via small, but coordinated conformational changes within core histones that help AUF bind.

F. Points that could be addressed by the authors include the following: (1) This reader is not entirely convinced of the reasoning behind the discrepancy between the ICP-MS measurement of RAPTA-T/AUF binding and that seen in the structure. Is it at least possible that other AUF binding sites are present, but simply unoccupied in the present structure? If so, this should at least be alluded to in the text. (2) Is there a possible way to show a negative control for this reported synergistic binding within the nucleosome? For instance, using site specific cross-linking or mutagenesis to form a "road-block" that would not allow the induced conformational changes from RAPTA-T binding that are reported to travel to the AUF site via changes as shown in Figs. 4-6? Simply put, can the synergism be hindered in a way other than effecting initial RAPTA-T binding? At some point along the way between the differing drug binding sites. (3) The term allostery is used in the title, but then really not addressed formally in the text. For example, there are numerous models for "true" allostery that are not examined, here. This is a common and reoccurring issue in the literature that may be splitting hairs, but at this point, synergy is probably the more appropriate term. (4) last point is a small one where this reader would like a more clear bit of discussion or even speculation on how these drugs binding to the NCP actually kill cancer cells better when together, as they clearly do from Fig. 1.

G. References seem appropriate except in the 3rd paragraph of the introduction where discussing other "well-defined allosteric drug mechanisms..." After listing many, no references to any of this past work were included.

H. The manuscript was reported in a clear and relatively concise manner by the authors.

Reviewer #3 (Remarks to the Author):

The work of Adhireksan et al. presents an intriguing study of drug cross-talk mediated by nucleosome rearrangement. Very briefly, the work aims to demonstrate that one drug changes the conformation of the nucleosome to allocate the second drug, explaining why the use of two drugs has an enhanced effect on cancer treatment.

As discussed with the Editor, unfortunately I am not in the position of evaluating the overall quality of the work, as I do not have experience in x-ray crystallography, and the manuscript bases most of its conclusion on x-ray experiments. However, I am glad to provide a feedback on the ICP-MS experiment the authors performed.

ICP-MS is a technique where the sample is sprayed through a plasma torch, mostly made of argon at extremely high temperatures (>10,000 K). This device completely destroys the analyte to the atomic level, and single atoms are quantified by MS; their molecular weight determines the type of atoms, while the intensity of the signal can be correlated with the number of molecules.

In this work, the authors exploit the fact that the two drugs contain rare metals (ruthenium and gold), so by quantifying the levels of the metals in purified chromatin they can judge how much drug was incorporated. This part seems properly done; those two metals are very efficiently detectable by ICP-MS (table with detection limits: <http://crustal.usgs.gov/laboratories/icpms/intro.html>). Moreover, their mass is pretty unique, meaning that it is very unlikely that they detected an interference (e.g. iron has the same mass of argon+oxygen, a common interference considering the plasma torch).

In conclusion, my evaluation is that the ICP-MS experiment was properly performed. I hope this

comment can be of help to evaluate the manuscript.

Reviewer #4 (Remarks to the Author):

In their article "An Allosteric Mechanism in Chromatin from a Drug-Drug Synergy" the authors identify a novel synergy between the chemo agents RAPTA-T and auranofin and demonstrate that RAPTA-T sensitizes the cells to auranofin's cytotoxic effects. They next show RAPTA-T allows auranofin to accumulate in chromatin using mass spectrometry. The authors found the presence of aurofin derived gold adducts in nucleosome crystals only in the presence of RAPTA, strongly suggesting an allosteric effect. Indeed, the authors go on to demonstrate using crystallography that upon binding RAPTA the conformational properties of the nucleosome itself change, allowing binding of gold particles.

Necessary revisions:

- The source of Aurofin's toxicity, and whether this toxicity is dependent on its binding the nucleosome is unclear. Further discussion of this in both the introduction and discussion would benefit a reader. The link between the drugs' synergy and why the authors examine nucleosome adducts needs to be made more clearly
- The 'acid patch' in the crystal structure depicted in Figure 2 could be delineated more clearly.
- Also in figure two, the significance of the blue and red shaded electrostatic potentials needs to be noted in the figure legend.
- The correlations modeled in Figure six would suggest amino acid residues important in the synergistic effect. Can these be mutated to demonstrate their importance?
- The panels in figure five should be separately labeled A, B, C, etc.

RESPONSES TO REVIEWER COMMENTS

We thank the four reviewers wholeheartedly for their valuable time and input on our manuscript. We have considered all of the recommendations of the reviewers, whose insightful comments have been extremely helpful in compiling this revised version. Below we outline our responses to the specific points of each reviewer in turn.

In response to comments by Reviewer #1:

1. *Given that the crystal structures have multiple ligands bound to the NCP, and that the crystallographic conditions have significantly higher drug concentrations than the in vitro experiments, there is a strong possibility that the crystal structures have more drug molecules bound to the NCP than is biologically relevant. Therefore, it's not clear that the effects observed in the MD simulations are representative of the in vivo NCP dynamics. This is especially worrisome for the RAPTA-T molecules, as the simulations have two drug molecules in very close proximity to one another, but if only one is bound at physiologically relevant concentrations then the NCP dynamics are likely to be significantly different.*

To address this important point, we conducted an additional analysis of the RAPTA-T/AUF–NCP system, but with the RAPTA-T adduct at site RU2 removed. An ~600 ns MD simulation of this RU1-only system shows that it behaves qualitatively identical to the dual RU1/RU2 system, giving rise to effectively the same alterations in the structure, conformational dynamics and structural couplings. This is described now in a new paragraph (2nd) in the Results section of “Allosteric Mechanism for Promoting AUF Adduct Formation”, with an additional figure included in the SI (Suppl. Fig. 12), and we have made some additional comments on this aspect throughout the Discussion section.

2. *Also, for AUF the experimental IC50 is four orders of magnitude lower than the concentration used in the crystallographic medium, again suggesting that in vivo only one AUF may be bound to the NCP and not the two observed in the crystal structure. The authors never address these possibilities. This manuscript needs to include evidence and a strong justification that the most biologically relevant state includes four ligand molecules, with two RAPTA-T molecules on one face of the NCP and one AUF molecules on each face.*

Indeed this may well be the case, and we have accordingly now addressed this point in detail in the Discussion section (5th paragraph). We also note that for AUF, as well as for RAPTA-T, a single adduct would be sufficient to interfere with nucleosome-nuclear factor interactions. We discuss these possibilities for biological activity and impact in a new paragraph (6th) in the Discussion section.

3. *The overall results are very exciting, however the authors should make an effort to examine their work in the wider scope of the field. For example, many models for allosteric regulation have been developed; how does the mechanism they observe fit in with these models?*

We cite the recent work of Kornev and Taylor (2015), who have proposed two fundamentally distinct allosteric mechanisms that can be in operation. We discuss our findings in light of these models in the Discussion section, in a new paragraph (4th).

4. *In addition, the phenomenon of histone-mediated allosteric pathways as a mechanism of epigenetic regulation has recently been proposed and extensively examined in MD simulations in the following paper:*

"Effects of MacroH2A and H2A. Z on Nucleosome Dynamics as Elucidated by Molecular Dynamics Simulations," S Bowerman, J Wereszczynski, Biophysical journal 110, 327

The authors should discuss how their study fits in with the results of this manuscript.

We have included a discussion of this very interesting and relevant study in the new paragraph 3 of the Discussion section. Additional citation of the work appears in the closing statements of the Discussion.

5. *The authors used ICP-MS to determine that RAPTA-T and AUF accumulate on chromatin fibers, and they then make the jump to saying this means they bind to the NCP. Its possible there are higher affinity binding locations for the drug molecules at other sites in chromatin, such as in linker DNA, linker histones, or other chromatin associated proteins, and that at biologically relevant concentrations significant fraction of these drugs bind to these alternative sites. This is especially problematic for AUF molecules, for which the only evidence of direct NCP binding is the crystal structure that was solved at AUF concentrations significantly higher than the IC50 for this molecule. The authors should discuss these possibilities.*

We have added a discussion of this possibility in paragraph 5 of the Discussion section.

6. *The methods section contains a large amount of important information and should be moved to the main text.*

The methods section has been relocated to the main body of the manuscript.

7. *From the methods section it is unclear what the exact protocol was for the MD simulations. Which simulations had a 1 fs timestep, and which had a 1.5 fs timestep? Exactly how long were the simulations, how much was removed for equilibration, and how much was used for each analysis method? Is the equilibration time removed from the plots in Figures 4 and S5? Giving approximate times isn't sufficient for these details.*

The technical details of our MD simulations are now described in the revised version of the Methods section ("Classical Molecular Dynamics Simulations" heading). Specifically, a time step of 1 fs has been employed during the thermalization phase and for the initial ~8 ns of MD. Then MD simulations have been performed using a time step of 1.5 fs, following the same protocol as employed in our previous studies on ruthenium anticancer agents binding to the NCP (Adhireksan et al., *Nat. Commun.* 2014, 5, 3462) as well as to naked DNA (Ma et al. *Angew. Chem. Int. Ed.* 2016, 128, 7441; Gossens et al. *J. Am. Chem. Soc.* 2008, 130, 10921).

Approximately 750 ns of MD have been performed for the RAPTA-T/AUF–NCP and RAPTA-T–NCP systems, ~700 ns of MD have been collected for the RAPTA-T/AUF and ~650 ns for the native NCP. This latter simulation has now been prolonged reaching ~700 ns of MD. For all systems, the first ~100 ns of MD simulations have been removed and analyses have been performed over the remaining trajectories. The results reported here refer to the subsequent ~600 ns of MD, for each simulated system. Figures 4, and Supplementary Figures 5 and 6 have now been updated, including the results from the prolonged MD simulations of the native NCP.

8. *The authors should make a convincing case that their results are reproducible and not the benevolent result of a single simulation trajectory.*

As described above in response to point #1, we conducted an additional analysis of the RAPTA-T/AUF–NCP system, but with the RAPTA-T adduct at site RU2 removed. An ~600 ns MD simulation of this RU1-only system shows that it behaves qualitatively identical to the dual RU1/RU2 system, giving rise to effectively the same alterations in the structure, conformational dynamics and structural couplings. This is described now in a new paragraph (2nd) in the Results section of “Allosteric Mechanism for Promoting AUF Adduct Formation”, with an additional figure included in the SI (Suppl. Fig. 12), and we have made some additional comments on this aspect throughout the Discussion section.

Therefore collectively, there is strong consistency in our simulation results since we find effectively the same discrete dynamic transitions occurring in any of the three NCP systems containing RAPTA-T adducts (RAPTA-T–NCP, RAPTA-T/AUF–NCP and RAPTA-T/AUF–NCP with only a single RAPTA-T adduct at the RU1 site), and we do not observe these conformational features in either of the systems lacking RAPTA-T adducts (native NCP and AUF–NCP). We have included these arguments in the Discussion section (3rd paragraph).

9. *The PDBs 5DNM and 5DNN should be made available to reviewers.*

The two structures have been uploaded as supplementary data.

10. *Why were only the last 200 ns of the simulations used for the cross correlation analysis? How reproducible are these analyses on other portions of their trajectories?*

We had included a figure (Suppl. Fig. 8) with the cross correlation matrices referring to the last ~200 ns of MD for each system, which allows the precise comparison among four different MD runs. These cross correlations are representative and reproducible during the MD runs as demonstrated by computing the cross-correlation matrices over different time windows. Specifically, the time windows considered for comparison refer to: (i) the last 200 ns of MD; (ii) the last 400 ns of MD and (iii) the entire production run, comprising 600 ns of MD. As a convergence test, the full matrices of the square deviations of each matrix element $[A(C_{ij})]^2$ and the overall root mean square deviations (RMSD) have been computed. The RMSD value ranges are always <0.02 indicating that there are no significant differences between the cross correlation matrices calculated over the three time windows. These results indicate that MD data are robust and reproducible within the simulated trajectories and are

now described in the Methods (“Cross-Correlation Analysis” section) with inclusion of new Supplementary Figure 9.

11. *It would be helpful if the authors provided tables of the cross-correlation values between the key NCP locations and drug molecules. How strong are the correlations between the four drug molecules? These are difficult to read from the plots shown in Figure S8.*

We have added the new Supplementary Figure 10, wherein the cross-correlation coefficients between the key NCP locations as well as the drug molecules are reported.

12. *Figure S7 is difficult to interpret without error bars. Are these results statistically significant?*

Full data and the associated statistical analyses have now been added to the SI as Supplementary Tables 4 and 5. The differences in the statistical occurrence for the close contacts, which average 8.6%, are highly significant, with an average standard deviation of only 1.4%. This is now noted in Supplementary Figure 7 as well.

13. *The authors should provide some discussion about possible mechanisms of action for these drugs. Why would binding at these sites on the NCP result in cell death?*

We have included a discussion and proposal of several possibilities for the cytotoxic effect of nucleosomal adducts in paragraph 6 of the Discussion section. We have also added three new figures, Fig. 7 and Supplementary Fig. 13 and 14, in support of these arguments.

In response to comments by Reviewer #2:

14. *This reader is not entirely convinced of the reasoning behind the discrepancy between the ICP-MS measurement of RAPTA-T/AUF binding and that seen in the structure. Is it at least possible that other AUF binding sites are present, but simply unoccupied in the present structure? If so, this should at least be alluded to in the text.*

Yes, this is indeed a possibility, as there are other potential sites outside of the nucleosome core regions in chromatin. We have added a discussion of this in paragraph 5 of the Discussion section.

15. *Is there a possible way to show a negative control for this reported synergistic binding within the nucleosome? For instance, using site specific cross-linking or mutagenesis to form a "road-block" that would not allow the induced conformational changes from RAPTA-T binding that are reported to travel to the AUF site via changes as shown in Figs. 4-6? Simply put, can the synergism be hindered in a way other than effecting initial RAPTA-T binding? At some point along the way between the differing drug binding sites.*

This is an interesting idea, and we plan on contacting a potential collaborator to work with us on such a study, but we feel it is really another story.

16. *The term allostery is used in the title, but then really not addressed formally in the text. For example, there are numerous models for "true" allostery that are not examined, here. This is a common and reoccurring issue in the literature that may be splitting hairs, but at this point, synergy is probably the more appropriate term.*

Reviewer #1 brought to our attention a very recent MD simulation-based study showing that there are allosteric networks within the nucleosome core, which can be modulated by histone variant changes (S. Bowerman & J. Wereszczynski, 2016). We have included a discussion of the findings of this study in relation to ours in the new paragraph 3 of the Discussion section. We also cite the recent work of Kornev and Taylor (2015), who have proposed two fundamentally distinct allosteric mechanisms that can be in operation. We discuss our findings in light of these models in the Discussion section, in the subsequent new paragraph (4th).

17. *Last point is a small one where this reader would like a more clear bit of discussion or even speculation on how these drugs binding to the NCP actually kill cancer cells better when together, as they clearly do from Fig. 1.*

We have included a discussion and proposal of several possibilities for the cytotoxic effect of nucleosomal adducts in paragraph 6 of the Discussion section. We have also added three new figures, Fig. 7 and Supplementary Fig. 13 and 14, in support of these arguments.

18. *References seem appropriate except in the 3rd paragraph of the introduction where discussing other "well-defined allosteric drug mechanisms..." After listing many, no references to any of this past work were included.*

We have added 4 supporting references for these statements (Introduction, 3rd paragraph).

In response to comments by Reviewer #3:

No revisions were requested.

In response to comments by Reviewer #4:

19. *The source of Aurofin's toxicity, and whether this toxicity is dependent on its binding the nucleosome is unclear. Further discussion of this in both the introduction and discussion would benefit a reader. The link between the drugs' synergy and why the authors examine nucleosome adducts needs to be made more clearly*

We have added a new paragraph at the end of the Introduction, where we elaborate further on the cytotoxic properties of AUF and our motivation for studying the nucleosome binding activity of this agent. We have additionally included a discussion and proposal of several possibilities for the cytotoxic effect of nucleosomal adducts in paragraph 6 of the Discussion section. We have also added three new figures, Fig. 7 and Supplementary Fig. 13 and 14, in support of these arguments.

20. *The 'acid patch' in the crystal structure depicted in Figure 2 could be delineated more clearly.*

The acidic patch has been more clearly described and designated for Figure 2. In addition, we have included a further illustration of this region in panel a of the newly added Figure 7.

21. *Also in figure two, the significance of the blue and red shaded electrostatic potentials needs to be noted in the figure legend.*

This has now been noted in the legend of Figure 2.

22. *The correlations modeled in Figure six would suggest amino acid residues important in the synergistic effect. Can these be mutated to demonstrate their importance?*

This is an interesting idea, and we plan on contacting a potential collaborator to work with us on such a study, but we feel it is really another story.

23. *The panels in figure five should be separately labeled A, B, C, etc.*

We have divided Figure 5 into four different panels now, with separate descriptions for labels a/b/c/d.

REVIEWERS' COMMENTS:

Reviewer #1 (Remarks to the Author):

The authors have nicely addressed my concerns, and I can now recommend publication in Nature Communications.

Reviewer #2 (Remarks to the Author):

For this reviewer, the story is potentially very exciting, but I am not sure the data paints a complete picture. Some of the recommended additional experiments or possible other interpretations were heard by the authors and some points addressed as discussion, but not clarified through meaningful new experiments (outside of MD). The data seems sound and is presented in a clear manner, yet is not totally convincing of the title or main points of interest.

Reviewer #4 (Remarks to the Author):

The authors have addressed my previous comments in a satisfactory manner.

RESPONSES TO REVIEWER COMMENTS

We wish to thank the four reviewers wholeheartedly once more for their valuable time and input on our manuscript.

In response to comment by Reviewer #2:

For this reviewer, the story is potentially very exciting, but I am not sure the data paints a complete picture. Some of the recommended additional experiments or possible other interpretations were heard by the authors and some points addressed as discussion, but not clarified through meaningful new experiments (outside of MD). The data seems sound and is presented in a clear manner, yet is not totally convincing of the title or main points of interest.

We feel that the additional experimental work would entail a new study, but we have accordingly revised the title and modified some of the key arguments in the text.